HIV-positive parents, HIV-positive children, and HIV-negative children’s perspectives on disclosure of a parent’s and child’s illness in Kenya

Gachanja Grace 1 g_gachanja@hotmail.com
Burkholder Gary J. 1 2 3
Ferraro Aimee 1
1 College of Health Sciences, Walden University , Minneapolis, MN , USA
2 College of Social and Behavioral Sciences, Walden University , Minneapolis, MN , USA
3 Laureate Education, Inc. , Baltimore, MD , USA
Crandall Keith
Electronic publication date: 2014 Jul 10
Publication date: 2014
Volume: 2
Electronic Location ID: e486
Received 2014 Mar 23; Accepted 2014 Jun 25
Copyright: © 2014 Gachanja et al.
Copyright year: 2014
Copyright holder: Gachanja et al.
License: This is an open access article distributed under the terms of the Creative Commons Attribution License, which permits unrestricted use, distribution, reproduction and adaptation in any medium and for any purpose provided that it is properly attributed. For attribution, the original author(s), title, publication source (PeerJ) and either DOI or URL of the article must be cited.
License URL: https://creativecommons.org/licenses/by/4.0/

Keywords: HIV/AIDS, HIV disclosure, Resource-poor nation, Child HIV status disclosure, Parental HIV status disclosure, Qualitative research, Kenya

Funding: No funding was received for this study.

==============================
HIV disclosure from parent to child is complex and challenging to HIV-positive parents and healthcare professionals. The purpose of the study was to understand the lived experiences of HIV-positive parents and their children during the disclosure process in Kenya. Sixteen HIV-positive parents, seven HIV-positive children, and five HIV-negative children completed semistructured, in-depth interviews. Data were analyzed using the Van Kaam method; NVivo 8 software was used to assist data analysis. We present data on the process of disclosure based on how participants recommended full disclosure be approached to HIV-positive and negative children. Participants recommended disclosure as a process starting at five years with full disclosure delivered at 10 years when the child was capable of understanding the illness, or by 14 years when the child was mature enough to receive the news if full disclosure had not been conducted earlier. Important considerations at the time of full disclosure included the parent’s and/or child’s health statuses, number of infected family members’ illnesses to be disclosed to the child, child’s maturity and understanding level, and the person best suited to deliver full disclosure to the child. The results also revealed it was important to address important life events such as taking a national school examination during disclosure planning and delivery. Recommendations are made for inclusion into HIV disclosure guidelines, manuals, and programs in resource-poor nations with high HIV prevalence.

Introduction

By 2012, over 35 million persons were infected with human immunodeficiency virus (HIV) globally (UNAIDS, 2013). Resource-poor nations, especially those within Sub-Saharan African (SSA) bear the brunt of the pandemic; 90% of infected persons live in SSA. Kenya has 1.4 million adults and 200,000 children living with the disease (NACC and NASCOP, 2012). As of 2011, the prevalence of the disease for adults aged 15–64 years was 5.6%, and that of children 18 months to 14 years was 0.9% (National AIDS and STI Control Programme, 2013). Due to increased access to antiretroviral treatment (ART), prevalence is expected to increase as infected persons live longer, thereby making control of the disease a national priority in the decades to come (NACC and NASCOP, 2012).

Disclosure of HIV status to children is important for HIV-positive parents, but many are extremely challenged by the disclosure process (Blasini et al., 2004; Delaney, Serovich & Lim, 2008; Kallem et al., 2011; Kennedy et al., 2010; Kouyoumdjian, Meyers & Mtshizana, 2005; Menon et al., 2007; Vallerand et al., 2005; Vaz et al., 2008). HIV disclosure is a process that moves a child from a state of nondisclosure to partial and then to full disclosure of illness (Wiener et al., 2007). Nondisclosure of illness has been defined as a state in which the child has no knowledge of the illness (Bikaako-Kajura et al., 2006; Kallem et al., 2011; Oberdorfer et al., 2006; Vaz et al., 2011). In partial disclosure the child has limited details (for example, awareness of daily medication consumption and presence of chronic illness) of the illness (Bikaako-Kajura et al., 2006; Rochat, Mkwanazi & Bland, 2013; Vaz et al., 2011). A child with full disclosure is aware that the illness in question is HIV/AIDS (Bikaako-Kajura et al., 2006; Kallem et al., 2011; Rochat, Mkwanazi & Bland, 2013; Oberdorfer et al., 2006).

HIV disclosure studies in SSA have centered on telling infected children about their own HIV statuses (Bikaako-Kajura et al., 2006; Brown et al., 2011; Kallem et al., 2011; Menon et al., 2007; Moodley et al., 2006; Vaz et al., 2010; Vreeman et al., 2014); a few have focused on telling children about their parents’ HIV statuses (Madiba & Matlala, 2012; Nam et al., 2009; Palin et al., 2009; Rochat, Mkwanazi & Bland, 2013; Rwemisisi et al., 2008). Recent studies on disclosure to HIV-positive children have revealed HIV disclosure rates between 11% and 26% in Kenya (John-Stewart et al., 2013; Turissini et al., 2013; Vreeman et al., 2014) and between 17% and 40% in other SSA countries (Biadgilign et al., 2011; Bikaako-Kajura et al., 2006; Fetzer et al., 2011; Kallem et al., 2011; Madiba, 2012; Menon et al., 2007). Research suggests that the typical age ranges of HIV-positive children receiving full disclosure of their own HIV statuses was between 3 and 19 years in Kenya (Vreeman et al., 2014); 8 and 18 years in the Democratic Republic of Congo (Vaz et al., 2010; Vaz et al., 2008); and 5 and 15 years in South Africa (Heeren et al., 2012; Moodley et al., 2006).

Rates of parental HIV infection status disclosure to children are slightly higher than those of disclosure of the child’s infection status. Recent studies conducted within SSA have revealed such disclosure rates ranging from 29% to 50% (Nam et al., 2009; Palin et al., 2009; Rwemisisi et al., 2008). The age ranges of children receiving full disclosure of their parents’ HIV statuses were between 9 and 17 years in Botswana (Nam et al., 2009) and 10 and 18 years in Uganda (Rwemisisi et al., 2008). We found only one study in SSA that examined the process of disclosure of a parent’s HIV status to their HIV-negative children (Rochat, Mkwanazi & Bland, 2013). Therefore, the knowledge on how parents approach and perform disclosure of their own HIV statuses as well as that of their children to their HIV infected and uninfected children within the same household is relatively undeveloped in the literature.

Three key theories have been published to help guide the HIV disclosure process. The disease progression theory stipulates that infected persons are prompted to fully disclose when their illnesses progress because they cannot hide their symptoms, ART consumption, hospitalizations, or the use of HIV support services (Serovich, 2001; Qiao, Li & Stanton, 2013). The consequence theory of HIV disclosure expounds further by explaining that after disease progression, the infected person weighs the benefits of disclosure versus the consequences that might occur due to such disclosure, proceeding only when the benefits outweigh the risks (Serovich, 2001; Qiao, Li & Stanton, 2013). According to the disclosure process model (Chaudoir, Fisher & Simoni, 2011; Qiao, Li & Stanton, 2013), the ultimate goal of full disclosure is to limit negative outcomes (rejection, poor relationships) while maximizing positive outcomes for both the parent (social support from children) and child (improved psychological and physical wellbeing). The purpose of our study was to understand the lived experiences of HIV-positive parents and their HIV-positive and negative children before, during, and after the disclosure process in Kenya.

Methods

Sample selection

This study was conducted at the HIV Comprehensive Care Clinic (CCC) located at the Kenyatta National Hospital (KNH) which is near the center of Nairobi, Kenya. HIV-positive parents and children were purposively recruited by the lead author from the waiting areas of the CCC; participants were also referred by clinical officers (equivalent of physician assistants in the U.S.), nurses, and peer educators from the examination and triage rooms during their regularly scheduled clinic visits. Additionally, a different group of HIV-positive parents were requested to nominate and bring their HIV-negative children at a time of their choosing to the clinic for participation in the study. Selected parents and children were not parent–child dyads because we aimed to obtain rich data representing the perspectives of parents and children from different families. Once referred by healthcare professionals (HCPs), the first author met with and informed potential participants about the research and its procedures through the informed consent process. She then escorted those who agreed to participate to a private room within the clinic where informed consent was obtained and study procedures performed.

We purposely selected HIV-positive parents having biological HIV-positive and negative children between the ages of 8 and 17 at various stages of disclosure (no, partial, full) of their parents’ and/or child’s own illness. To prevent inadvertent disclosure during the interviews we selected HIV-positive children with partial and full disclosure of their own illnesses; and HIV-negative children with partial and full disclosure of their parents’ illnesses. Children with partial disclosure of illness were interviewed based on what illness (for example, tuberculosis or backache) those children knew they themselves or their parents were suffering from. Purposeful selection of participants was conducted over a six-week period; interviews were conducted until saturation was reached.

Ethics approval was received from the KNH Research Standards and Ethics Committee (Approval # P373/10/2010), and the Walden University Institutional Review Board (Approval # 11-10-10-03904). HIV-positive and negative children provided written assent to participate in the study and their parents provided written informed consent. HIV-positive parents also provided written informed consent. All participants consented/assented to the digital recording of their interviews. Due to the sensitive nature of the research topic, the CCC psychologist remained available throughout the data collection period to provide counseling if needed following their interviews; one HIV-negative child was provided with counseling services. Participants were also invited to visit the psychologist’s office after study participation if they wanted to speak about their experience.

Data collection and instruments

Interpretative qualitative data were collected using in-depth semi-structured interview guides adapted from those used by Vaz et al. (2008). We adapted (with permission) the guides by removing questions that did not specifically address disclosure. The adapted guides were reviewed and approved by members of the research team (GB and AF) familiar with qualitative instrument design. Interviews were performed in English (the official language of Kenya) by the first author in a private room at the CCC and lasted from 30 to 90 minutes. All selected participants were fluent in English. Parents of HIV-positive and negative child participants were given the option of being in the room while their children were being interviewed but all declined; all child participants assented to being interviewed alone.

HIV-positive parents’ interview guide questions explored various aspects of the disclosure process including reasons for and against full disclosure, parents’ activities conducted in preparation for full disclosure, the family’s needs during the disclosure process, when and how partial and/or full disclosure was performed to their children, their recommendations on how full disclosure to children should be approached, and what were the anticipated/actual reactions and consequences of full disclosure. HIV-positive and negative children’s interview guide questions explored when and how they had received partial or full disclosure of their own and parents’ illnesses respectively, how they felt during and after partial or full disclosure of their own and parents’ illnesses respectively, and their recommendations on how full disclosure should be performed to children.

Data analysis

Recorded interviews were immediately transcribed by the first author and a local university student trained in transcription. Transcripts were rechecked against the interviews for accuracy; two transcripts needed minor editing corrections. Five parents who had provided email addresses were also sent their transcripts to verify the accuracy of the transcription, and they verified the accounts of their interviews as being accurate. Transcripts were uploaded into NVivo 8 for management and analysis. Data analysis was performed using the modified Van Kaam method (Moustakas, 1994) by the first author and cross-checked (by GB and AF). Analysis included thorough reading of transcripts, listing and grouping those with similar information while checking for codes, then clustering codes into emerging themes combined with predetermined themes (age of disclosure, child maturity and understanding level, person to perform disclosure) from prior research. Over 300 total codes emerged from the data, and these were crosschecked (by GB and AF) against a select group of transcripts. A theme with six subthemes detailing the disclosure process emerged as described below.

Results

Sample description

Sixteen HIV-positive parents, seven HIV-positive children, and five HIV-negative children were recruited into the study. The parents ranged in age from 30 to 54 years of age; 11 were women, the higher percentage of women was expected, as it is mothers who typically bring their children to clinic. One married couple was interviewed to obtain a husband’s and wife’s perspective on disclosure within the same family. Ten parents had 1–2 children, three had 3–4 children, and the last three (including the married couple) had 5–6 children. We sought children between the ages of 8 and 17 years; however, children aged 12 years and older who had already received partial or full disclosure of their own or their parents’ illnesses comprised the children’s sample. Selected HIV-positive children were between 12 and 17 years of age. For six children, parents had fully disclosed these children’s illnesses to them, and one was only aware he was suffering from TB. HIV-negative children were between 12 and 16 years of age; three had full disclosure of their parents’ illnesses, and two were sisters who were only aware their mother was suffering from chronic backache. The study participants’ genders and HIV statuses are displayed in Fig. 1 and their social demographic profiles are presented in Table 1.

Figure 1 Sample description.

FD, full disclosure; ND, nondisclosure; PD, partial disclosure.

Table 1 Participants’ social demographic characteristics.

Variable	HIV-positive
parents	HIV-positive
children	HIV-negative
children	
Age				
Children 12–13		2	1	
14–15		1	3	
16–17		4	1	
Parents 31–40	8			
41–50	7			
51–60	1			
Gender				
Female	11	3	3	
Male	5	4	2	
Educational status				
Primary	2	2	3	
Secondary	7	5	2	
College	7	NA	NA	
Marital status		NA	NA	
Single	1			
Divorced	1			
Widowed	4			
Married	10			
Notes.

NA Not applicable

The perceptions of the participants regarding how full disclosure should be delivered to children are presented in the six subthemes below and displayed in Fig. 2. These subthemes include considering as a process the parent’s right to disclose and the child’s right to receive disclosure. Issues needing to be addressed during the disclosure process included full disclosure of multiple HIV infected family members’ (parents and children) illnesses, the parent’s and child’s health statuses, child’s understanding and maturity level, and the person who should fully disclose to children are further discussed.

Figure 2 HIV disclosure subthemes.

Disclosure is a child’s right and a parent’s decision

All parents were willing at some point to fully disclose to their children; some felt it was the right of the child to know about the parent’s and/or child’s illnesses. Parents expressed they needed help from healthcare professionals (HCPs) to be able to fully disclose to their children. However, with the exception of one parent who advised parents perform disclosure without heeding from HCPs, a few were completely against being told by HCPs when to fully disclose to their children. Parents wanted HCPs’ advice on how to proceed with disclosure and then be given adequate time to prepare and fully disclose at a time of their choosing:

The question of passing information is dependent on each and every family, the time they find is best suited for them. Even if they are talking to a counselor or anybody else, they should be encouraged to get the most suitable time they think is important for them to pass on the information to the child because they are the ones who know their child because they need from them personal information that causes a lot of repercussions good or bad.

54-year-old father of two HIV-negative children and three untested children

Both HIV-positive and negative children agreed with parents that they needed to know about the illness. HIV-positive children wanted to be told about their illnesses because once they became fully aware of their health statuses, they could more easily deal with the illness:

Children should know because somehow we take care of ourselves, you know now we eat some fruits to boost our immune system, they should know.

13-year-old HIV-positive boy

At least I know so that if I get into a relationship without knowing I can infect the person, and so it is not good. So at least they told me earlier that I will take the measures not to make other people get infected so that we can eradicate AIDS for good.

17-year-old HIV-positive girl

The three HIV-negative children with full disclosure of their parents’ illnesses were curious about how their parents became infected but were happy to know because secrets were eliminated and they realized the disease was real. HIV-negative children were highly in favor of children being informed of their parents’ illnesses followed by subsequent testing of all children:

It is better for them [children] to know because this is what we live for, to know about our parents, about ourselves, about our family and to know our status is knowing your whole life, how it is going to be, how you are going to live it… If a parent hides his or her status from a child, let’s say he is doing bad behaviors or something, he might be having it and contracts it to other people… I was relieved to know that me too, I can get tested for a reason… I got tested at school, there were these people coming to test us at school. Then they tested us, then we got our results, I was negative.

15-year-old HIV-negative boy

Although parents thought children should know about the illness, they considered full disclosure to be very challenging, describing it as “weighty”, “hurting”, and a “burden”. Disclosure was even challenging to parents who had already fully disclosed to their older children and were in the process of preparing younger siblings. Parents were only willing to proceed with full disclosure when they and their children were ready. In the meantime they postponed full disclosure by telling their children the parent or child was suffering from another illness such as tuberculosis, backache, and high blood pressure to explain why there was daily consumption of medication.

Both HIV-positive and negative children were against being lied to about the illness by their parents because it betrayed their trust. They were all in favor of disclosure in a timely manner because at some point it would be inevitable for them to know the truth:

Even if they didn’t tell me sooner or later I would come to know.

16-year-old HIV-negative girl with partial disclosure of her mother’s illness

Children should be given a chance to know because if you lie to your child, when he or she grows up it becomes very difficult for you to relay the message to your child. So you must tell your child when he or she is little coz a little child is easier for you to handle but for a grownup it becomes very complicated.

17-year-old HIV-positive boy

Disclosure is a process

Parents expressed that full disclosure to a child be performed as a process. By disclosing in stages, children absorbed the news better, suffered less negative impact, and did not forget what they had been told. Some parents who had fully disclosed their children’s illnesses explained HIV-positive children needed to be reminded of their illnesses and to take ART:

You know you don’t tell them today and then you forget about it, keep on telling them the awareness so as not to forget their status and their medication… They [5 and 7 year-old HIV-positive children] know they are taking medicine for HIV but they don’t understand. Now we are waiting when they mature at 10, then we will tell them the problem you are taking medicine is this and you should not stop.

45-year-old HIV positive father of two HIV-positive children and a HIV-negative child

HIV-positive children agreed taking ART was tedious and that they were sometimes forgetful:

HIV is a non-curable disease and you have to take drugs each and every day and sometimes you forget and you become sick and many complicated issues.

17-year-old HIV-positive boy

HIV-positive and negative children also stated they preferred to receive disclosure over time. Some children who had received full disclosure without lengthy preparation expressed they had been impacted by the news:

There is a newspaper that we had read together, it was about this woman who had lived with HIV for 20 years, so we talked about it briefly. Then she [mother] came after finishing cooking, I think she asked me how if I knew she was positive what would I do? And I just told her that I wouldn’t do anything since she is the one who has taken care of us since the time we were young. So when she told me at first I was shocked… I really didn’t know, it was just a shock, and it came as a shock to me.

14-year-old HIV-negative girl

Disclosing multiple illnesses in the family

Many parents in the study had HIV-positive living (or dead) spouses and children. None of the HIV-positive children in the sample reported living HIV-positive siblings but some had siblings or a parent who had passed away. None of these HIV-positive children expressed they had been told the deaths were due to an AIDS-related illness; however, they voiced suspicions about it. HIV-negative children in the sample also had no HIV-positive siblings, but three had lost a parent due to a cause of death unknown to them. Children therefore had a need to know about the illnesses and causes of deaths within their families. A 17-year-old HIV-positive boy expressed he was surprised to learn of his illness because no one else in his family was infected except “maybe my mother, she passed away when I was three months”. A 12-year-old HIV-negative girl expressed she wanted her mother “to comfort me and tell me why father had died because he had no illness”.

HIV-positive parents in the sample had a mixture of children with no, partial, or full disclosure of illnesses within their households, and they performed preferential disclosures opting to fully disclose all illnesses in the family to older but not younger children. Incidences of full disclosure of a parent’s illness at the time of diagnosis and disclosure of an infected child’s illness were reported by three parents either because the child guessed the parent was similarly infected, or needed encouragement that they were not alone. Fully disclosing all illnesses in the family was cathartic for some parents:

Because it was very difficult for me to disclose I wanted him to know everything at once… One day he asked about his [late] father in the morning and I got that opportunity… I told him it was the chest due to HIV, he was HIV-positive… He said oh, no, no, no then that explains why I take this medication am I also HIV-positive? So I just said yes… He asked are you also sick? That is when I told him I am also sick… It was a relief for me I just told him everything and then we discussed and it was a bit lighter. In fact at the end of it all he said I will pray for you, we will take care of one another.

37-year-old HIV-positive mother of one HIV-positive boy

Some parents who had not yet fully disclosed planned to disclose all illnesses to their children at the same time:

I will try to do it in one session [disclose both parents’ illnesses] if she understands me and I see she understands the situation and accepts it, then I’ll do all of it in one session.

39-year-old father of a HIV-negative daughter with partial disclosure of the parents’ illnesses

Although cathartic for parents, full disclosure of many family members’ illnesses at the same time was traumatic for some children. One HIV-negative child was so upset recounting how he received full disclosure that his interview was stopped early and he was referred to the psychologist’s office for counseling:

He [father] told me that when I was five years my mother died and told me not to tell anyone about it, I felt sad (starts to cry). He told he was using some drugs and I never asked him more… He told me the current mother who took care of me until today I am still staying with her is my stepmother (sniffles).

14-year-old HIV-negative boy

Some child participants including children of HIV-positive parents, absorbed and overcame the shock of the news, and subsequently moved on with their lives:

I just take it to be normal because I cannot just hate her because she has got an illness. I see her every day, but nothing comes to me, even I watch television, I see people talking about AIDS, HIV how it kills. I don’t even remember if my mother has such a kind of illness, I just take her as normal and I am dedicated to taking care of her.

15-year-old HIV-negative boy

Other children, especially those who received full disclosure as teenagers, took longer to recover. A few parents whose older children received full disclosure poorly were hesitant to disclose to younger siblings, especially if these younger siblings were close to them or were known to have tempers:

I know she [second-born daughter] is hot tempered, so I was thinking maybe let us leave her because she is not even mature until she reached maybe I thought 17 there, I could tell her after she had finished her fourth form. Even the one who is following her is the same.

37-year-old mother of one HIV-positive child, one HIV-negative child, and two untested children

Considering a parent’s and a child’s health status

When considering full disclosure, parents took into account their and their children’s health statuses. Both parents and children cautioned even without full disclosure, children were highly aware of the disease and were able to guess the nature of illness:

If you have been in and out of hospital, you get very ill you are taken to Kenyatta Hospital then you are tested you come out you are very weak, they will even know before you tell them coz they might have already guessed, you are coughing, you are what.

54-year-old HIV-positive father of two HIV-negative children and three untested children

Some parents when they don’t tell you, the signs just show and the child doesn’t let’s say study well, he just gets to feeling so low.

15-year-old HIV-negative boy

Parents were unwilling to fully disclose their illnesses to their children when they (parents) were in poor health; however, they advised it was permissible in certain instances. These included when the parent was too sick to take him/herself to the hospital or pick up medications from the CCC. The child with full disclosure could take the parent to the hospital, pick up ART refills at the CCC, take care of siblings, and perform household tasks until the parent was feeling better. HIV-negative children provided responses similar to parents about being against full disclosure when a parent was unwell; and described vivid memories of the time their parents had been severely sick:

Father had died, she [mother] had an illness because she had carried a baby in her stomach, my last sister… She was weak, very weak that even she could just fall down… I thought that maybe she could pass away even her coz of the stress that she had. And with the stress how could she manage to get all the school fees for us all and feed the family.

12-year-old HIV-negative girl

Generally parents were of the opinion that HIV-positive children should not be told of their own illnesses when they were in poor health because it might make them lose hope and cause their condition to worsen. They therefore advised other parents to wait until both the parent and child were in good health prior to fully disclosing to a HIV-positive child:

Tell them when you [parent] have the strength because she might have that feeling why did my parents wait until they are bedridden, they can’t walk? You have to tell them early to prepare them psychologically. You also have to tell [HIV-positive] children when they have the strength, they are energetic, they know what they can do because sometimes if you tell them when they are bedridden it won’t help. Tell them when they have the energy then they will know what food they can eat, this medicine is for this.

45-year-old father of two HIV-positive children and one HIV-negative child

However, if the HIV-positive child’s health had deteriorated due to poor ART adherence then it was imperative he or she receive full disclosure:

He has to know why he is taking medicine and why he is getting ill because maybe when he is not taking the medicine proper, you have to tell him or her you have to take your medicine so that you can be well.

46-year-old mother of two HIV-positive children and one HIV-negative child

Child’s understanding and maturity level

Parents expressed that children developed and matured differently; thus, the decision to fully disclose should be assessed individually. Parents were especially hesitant to fully disclose to young children because they feared these children would not understand the meaning of the illness or keep the information a secret thereby exposing the family to stigma:

You know when they are young they will just take it like this… She [nine-year-old daughter] does not know how to keep a secret because she will shout to the other children, that’s why I fear to tell her.

32-year-old mother of one HIV-positive daughter and one untested daughter

Both HIV-positive and negative children agreed it was not wise to fully disclose to young children:

Even if you tell them [HIV-positive children] when they are small they cannot even know what they are supposed to do.

16-year-old HIV-positive girl

You cannot tell a child at seven that they have HIV, maybe he will not understand it or he will not handle it.

15-year-old HIV-negative boy

One HIV-positive child thought children should receive full disclosure by five years because when they sensed the information was important, they would always remember it. One parent had fully disclosed his, his wife’s, and two children’s illnesses to all his children when aged 3, 5, and 10 years but the two younger children did not comprehend what it meant and he wanted to re-inform them again at 10 years of age. Other parents and children disagreed with full disclosure at an early age preferring to provide limited information over time corresponding to what children were being taught in school about the disease:

When the child is about six years you cannot tell her or him that you are HIV-positive, but you can tell the child that I am on medication and then after two years you just disclose after 10 years. 10 years he has knowledge to know what is HIV.

46-year-old mother of two HIV-positive sons and one HIV-negative daughter

I am saying [start telling them at] seven because by the time they are eight years most of them are in class three and nowadays they teach at school, it is already in their books.

37-year-old mother of two HIV-positive children

Some parents reported children were initiating sex by 9 years and were therefore in need of full disclosure around this age. HIV-positive and negative children agreed teenagers were having sex mostly due to peer pressure and they needed to know of the illness so they would not infect others or become infected respectively. The most frequently cited age by both parents and children when full disclosure should be performed was 10 years because at this age, the child was capable of understanding the illness. The participants’ preferred ages of full disclosure to children are displayed in Table 2:

Table 2 Age at which parents should fully disclose to children.

	Child’s age at disclosure	
	5	9	10	11	12	13	14	15	17	
HIV-negative children (N = 4a)			1		2			1		
HIV-positive children (N = 7)	1		1			1	2	2		
HIV-positive parents (N = 14b,c)		1	5	1			4		3	
Total N = 25a,b,c	1	1	7	1	2	1	6	3	3	
Notes.

a One HIV-negative child did not complete the entire interview.

b One HIV-positive parent said when the child is mature.

c One HIV-positive parent said immediately ART is commenced.

When the parents feel comfortable from 10 and above, tell them [HIV-positive children] slowly about their illnesses.

13-year-old HIV-positive boy

Tell them [about a parent’s illness] when the child is young, maybe about 10 years because the child will have grown a bit.

12-year-old HIV-negative girl

You can tell a child whom you can tell and understand, like now you can’t tell a five-year-old. I think that age of between 10 and 13 because the child is a bit mature but since the way he said I should have told him earlier I feel that 10 is the best age.

37-year-old mother of one HIV-positive son who received full disclosure at 12 years

Both parents and children expressed children were mature enough to receive full disclosure by 14 years of age. At 14–15 years of age, most children are in Standard 8 during which they take their national primary school exit examination. Subsequently, the majority of children are admitted to a boarding secondary/high school. Some parents and children advised children be given the opportunity to take the examination first before being told of their and/or their parents’ illnesses:

I liked it because if she [mother] had told me a while back, I don’t know how I would have done my KCPE [Kenya Certificate of Primary Examination], so I’m just glad she told me after.

15-year-old HIV-negative girl who had just finished taking the KCPE at Standard 8

When I knew about my HIV status I was about to do my KCPE, I was in class eight, I fell so sick… The message really (pauses) intruded into my life and I was very depressed so it made me to fail my exams.

17-year-old HIV-positive boy

I was feeling the best time will be after he completes his class eight because that is the time we take them to a boarding school. So we feel it is good when the child knows because he will be carrying his own medicine.

39-year-old mother of one HIV-positive son and one untested son

Nine parents had fully disclosed their illnesses to at least one child in their household; and eight parents had fully disclosed a child’s illness to at least one of their HIV-positive children. Parents reported fully disclosing to their children when they were between the ages of 3 and 20 years. Six HIV-positive children received full disclosure of their own illnesses while between the ages of 13 and 17 years, and three HIV-negative children received full disclosure of their parents’ illnesses when they were 14 years of age. HIV-positive and negative children’s disclosure statuses are displayed in Table 3.

Table 3 HIV-positive and HIV-negative child participants’ disclosure statuses.

Child	Sex	Age at full
disclosure	Disclosure
status	Current
age	
HIV-positive children’s disclosure statuses of their own illnesses	
A	M	9	FD	13	
B	M	“Very young age”	FD	15	
C	F	13	FD	16	
D	F	14	FD	16	
E	M	14	FD	17	
F	F	17	FD	17	
G	M	NA	PD	12	
HIV-negative children’s disclosure statuses of their parents’ illnesses	
A	M	14	FD	14	
B	F	14	FD	14	
C	M	14	FD	15	
D	F	NA	PD	12	
E	F	NA	PD	16	
Notes.

F female

M male

FD full disclosure

NA not applicable

PD partial disclosure

Person to perform disclosure

Participants were asked to state who they thought was the best person to deliver full disclosure of illness to the child. Their responses are presented in Table 4. The majority of parents felt they were the ones who should fully disclose to their children. Married parents felt it was better for them to liaise and present the news to their children together with one taking the lead. Parents who had already fully disclosed advised other parents to disclose in a calm manner at a time when they could address all of the child’s questions:

Table 4 Person to perform disclosure.

	Person to perform disclosure	
	Parent	HCP	Relative	
HIV-negative children (N = 4a)	2		2	
HIV-positive children (N = 7)	3	4		
HIV-positive parents (N = 16)	14	1	1	
Total N = 27a	19	5	3	
Notes.

a One HIV-negative child did not complete entire interview.

You have to be straight to the point, you have to control yourself when you are saying this in order for them also to be calm. You have to be their pillar now you are saying you are sick they are seeing your emotions are controlled so they will sort of get the strength from you. But if you break down and also bring words that will make them think the situation is hopeless, now you also you have given up. Actually you have to choose your words but generally the choice of words don’t come in, it’s just how well you are managing your emotions in front of them.

49-year-old mother of two HIV-negative children and three untested children

Where parents felt incapable of fully disclosing on their own, they advised these parents have an HCP or a relative (for example an aunty or grandmother) close to the child be present as a support person:

It is the duty of the parent to tell their children they are HIV-positive maybe through the help of social workers, counselors, doctors or nurses to make their children understand what is HIV, coz HIV is a scare if you do not understand what it is. It can destroy a child’s life if the child does not understand what HIV is.

39-year-old father of one HIV-negative child

One parent whose son received full disclosure from a VCT counselor following testing cautioned against an HCP unfamiliar to the child delivering full disclosure:

From the way my son reacted, I think it is the parent [who should deliver full disclosure], that way whatever bad or silly question he might want to ask, he will be free to tell you than in the presence of a counselor… You see he was just crying then he was sad, even though we gave him time to ask questions or say whatever he wanted to say he couldn’t, this was a stranger he had never seen her, he was never close to the counselor. I think he didn’t want to say whatever he wanted to say, but if we could have been two, I am sure he could have said many things or asked whatever he wanted to ask.

37-year-old mother of two HIV-positive children

Half of HIV-negative children were in favor of their parents’ disclosing to them while the other half wanted a close relative (for example their older sister or aunty) to fully disclose to them. The majority of HIV-positive children thought full disclosure should be delivered to them from an HCP trained in disclosure:

Somebody who is not trained can come and tell you straight to the point. You know what you are HIV-positive and he or she cannot explain more about it. And then after all you hear about HIV it’s a killer disease, some people end up committing suicide and so many things, so I think a trained person is the one who should.

17-year-old HIV-positive boy

One parent who had fully disclosed to her child while alone at home agreed it was wise to have a trained HCP present at the time of full disclosure to an infected child:

I think what I should have done is at least disclose in the presence of somebody else who is aware like a counselor or something because immediately when he cried, I panicked, I really panicked because he really cried, he cried a lot and I was like now what do I do next, but I thank God I was able to control it.

37-year-old mother of one HIV-positive boy

Discussion

Our study revealed six subthemes associated with the HIV disclosure process. Parents did recognize it was a child’s right to receive full disclosure but viewed disclosure as a process that occurs over time; children agreed with them. When considering fully disclosing, it was important for parents to decide who was going to disclose, which family members’ illnesses and prior deaths to disclose, whether to postpone disclosure until the parent and/or child was healthy, and which child was capable of understanding or mature enough to receive the news. Overall, parents appear to favor disclosure at their own time without prompting from HCPs but acknowledged needing guidance during the process. Parents have reported requiring assistance from HCPs during disclosure preparation (Heeren et al., 2012; Nam et al., 2009; Nostlinger et al., 2004; Oberdorfer et al., 2006; Rwemisisi et al., 2008; Vallerand et al., 2005; Vaz et al., 2010; Vaz et al., 2008). Despite parents’ desire to delay full disclosure, both HIV-positive and negative children reported wanting to receive full disclosure of all illnesses and deaths in the family in a timely manner. Where many family member’s illnesses are to be fully disclosed to a child, intense predisclosure counseling should be provided to improve the child’s resiliency and capacity to absorb the news and also to assist the parent in delivering the news in an appropriate manner (Murphy & Marelich, 2008). Post disclosure, HCPs should assess the family to check how they are doing (Bikaako-Kajura et al., 2006; Murphy, Roberts & Hoffman, 2003).

HIV-positive parents are known to delay testing their children due to fear of obtaining another positive test result (Eisenhut et al., 2009; Ferrand et al., 2007; Ishikawa et al., 2010; Rwemisisi et al., 2008). In this study, we found that HIV-negative children supported testing of all children in that household; their statuses would be known and hopefully this would prevent HIV-negative children from becoming infected. HIV-positive children wanted to be informed of their illnesses so they could adhere to their ART, take better care of themselves, and gain self-independence. These needs expressed by HIV-positive and negative children should be a part of regular counseling provided to HIV-positive parents who seek guidance on full disclosure to their infected and uninfected children. Participants in this and another study (Heeren et al., 2012) recommended full disclosure as a process starting with limited information at 5 years of age. However, one parent in this study fully disclosed the parents’ and children’s illnesses to his 5 and 7-year-old children; the parent reported that the children did not understand what it meant. It therefore appears children should receive full disclosure at an age when they are capable of understanding. Many participants in the present study felt that this age is 10 years although few children at this age had full disclosure. Fourteen years was the age when most children received full disclosure of a parent’s and/or a child’s illness; this was the age when participants considered children mature enough to receive full disclosure if it had not been delivered before. The recommended age range of 10–14 years found in this study has been seen in prior studies conducted in resource-poor nations (Biadgilign et al., 2011; Moodley et al., 2006; Oberdorfer et al., 2006); and is in alignment with recommendations made by prior researchers that children receive full disclosure before they reach adolescence where more negative impacts have been noted (American Academy of Pediatrics, 1999; Blasini et al., 2004; Lester et al., 2002; Siripong et al., 2007). Full disclosure within this age range would educate children about the illness before sexual debut. Additionally, this study revealed that important life events such as completion of a national examination should be incorporated into full disclosure planning and delivery.

Parents knew children were starting sexual activity at 9 years and children confirmed they knew their teenage peers were having sex. Early sexual activity, many partners, and low condom use has been cited as some of the reasons why HIV spreads in Kenya (Republic of Kenya, 2009). This early onset of sexual activity among children is worrisome and has implications for the control of the disease within the country. The present study as well as one conducted in Botswana (Nam et al., 2009), revealed HIV-positive parents and their children are highly in need of sexual programs aimed at increasing parent–child communication on sexual matters, counteracting early sex initiation, reducing sexual-related peer pressure among children, and improving use of barrier methods during sex. These programs could prompt parents to fully disclose to their children during the recommended age period of 10–14 years if HCPs recommend testing of all children of newly diagnosed parents (Rwemisisi et al., 2008; Were et al., 2006).

The majority of parents in this study wanted to be the ones fully disclosing to their children, some alone and a few others with assistance from HCPs or close family members; the desire for full disclosure by the parents has been confirmed in prior research (Bikaako-Kajura et al., 2006; De Baets et al., 2008; Heeren et al., 2012; Kennedy et al., 2010; Kouyoumdjian, Meyers & Mtshizana, 2005; Nam et al., 2009). HIV-negative children agreed with parents, expressing they preferred to receive full disclosure from their parents or close family members such as older siblings or aunties. It is interesting to note that most HIV-positive children preferred to receive full disclosure from an HCP trained in disclosure delivery rather than a parent. As part of disclosure planning, an assessment may be necessary to decide who is the best person suited to fully disclose to a child (especially HIV-positive children), even where parents may want to be the ones delivering the news.

The results of this study do not appear to support the disease progression theory (Serovich, 2001; Qiao, Li & Stanton, 2013). Parents were unwilling to fully disclose when they or their children were in poor health. However, the results do support the consequence theory of HIV disclosure, as parents were willing to fully disclose their illnesses to children when the parents were very sick, so those children could support and take care of their parents (Serovich, 2001; Qiao, Li & Stanton, 2013). Parents were only willing to fully disclose to a sick HIV-positive child suspected of poor ART adherence so the child could take their medications well and improve his or her health. In these two aforementioned circumstances, the benefits of disclosure outweighed the consequences of nondisclosure making it wise to fully disclose (Serovich, 2001; Qiao, Li & Stanton, 2013). In support of the disclosure process model (Chaudoir, Fisher & Simoni, 2011; Qiao, Li & Stanton, 2013), both parents and children favored full disclosure delivered in an appropriate manner that ensured a good outcome for both parties. Training on HIV disclosure models is advocated for HCPs who work with HIV-positive families so they can better facilitate full disclosure delivery from parent to child and improve outcomes for all family members.

Qualitative research methodology focuses on revealing the meaning of sociocultural practices by exploring new complex problems from the point of view of a small group of participants (Creswell, 2009; Ulin, Robinson & Tolley, 2005). We interviewed a small group of purposely selected participants with the aim of understanding the lived experiences of HIV-positive parents and their children during the disclosure process in Kenya. We believe our sample of HIV-positive parents, HIV-positive children, and HIV-negative children provided us rich data that revealed the circumstances faced by HIV-positive parents and their children during the disclosure process. However, our sample presented three limitations. We interviewed more mothers than fathers; future researchers may want to focus on fathers’ experiences in the disclosure process. Parents and children in our study were from different families (we did not interview parent–child dyads), this also presents an important area of future research. Conducting interviews in English may have limited the participants to only those fluent in the language, future researchers may want to conduct interviews with those who only speak the native language or who are less fluent in English. Larger quantitative studies from a public health perspective are needed to determine the best way to fully disclose family members’ illnesses and deaths within heavily affected families. Additionally, the disclosure needs and preferences of HIV-positive and negative children need to be investigated further to see if they differ.

Conclusion

This article presents rich data on how HIV-positive parents approach full disclosure to their HIV-positive and negative children; and how full disclosure is received and perceived by HIV-positive and negative children. Prior studies have mostly focused on disclosure of a parent’s illness to their children or disclosure of their own illnesses to HIV-positive children. This study’s results are therefore important because they provide a thorough overview of full disclosure delivery from the point of view of HIV-positive parents; and disclosure reception from the viewpoint of HIV-positive and negative children. The recommendations provided by participants in this study help address the gaps in knowledge on disclosing many family members’ illnesses and deaths to children. Other important factors that should be considered include a parent’s and/or a child’s health status, children’s understanding and maturity level, important life events (for example national examinations), and the person best suited to fully disclose to a child. The recommendations from this study should be incorporated into preexisting or new guidelines, manuals, and programs on HIV disclosure from parent to child in resource-poor nations with high HIV prevalence.

We would like to thank Dr. Charles Kabetu and his office staff (Ms Mugambi and Ms Gacheru), Nelly Opiyo, Godfrey Mureithi, David Mutabari, and other KNH CCC staff for their help during this study. We would also like to thank Purity Kibino for her assistance with data transcription.

Additional Information and Declarations

Competing Interests

Author Contributions

Human Ethics

Gary J. Burkholder is an employee of Laureate Education, Inc.

Grace Gachanja conceived and designed the experiments, performed the experiments, analyzed the data, wrote the paper, prepared figures and/or tables, reviewed drafts of the paper.

Gary J. Burkholder and Aimee Ferraro analyzed the data, reviewed drafts of the paper.

The following information was supplied relating to ethical approvals (i.e., approving body and any reference numbers):

Ethics approval was received from the Kenyatta National Hospital Research Standards and Ethics Committee (Approval # P373/10/2010) in Nairobi Kenya, and the Walden University Institutional Review Board (Approval # 11-10-10-03904).

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
