# Peer review of "HIV-positive parents, HIV-positive children, and HIV-negative children’s perspectives on disclosure of a parent’s and child’s illness in Kenya"

_PeerJ, doi:10.7717/peerj.486_

## Round 0.1 · original submission · Major Revisions

Your paper has been now reviewed by two external reviewers (there is a third reviewer who agreed, but is very overdue at this point. If this review comes in, we will pass it along as well). Both reviewers feel your topic is important and your study appropriate for the journal. However, both also have a number of major issues for you to address before the paper can move forward. I concur with these evaluations. Please note Reviewer 1 has attached an annotated draft so see that attachment for detailed comments.

Reviewer 1 ·

Basic reporting

This is a very important topic and certainly addresses a gap in the literature. My greatest concern is that the qualitative research findings are being reported in comparison to quantitative research findings. The results and discussion section should be revised to reflect the strengths and limitations of qualitative research findings. Please see my comments in the document.

Experimental design

This qualitative study is very interesting. It is a well designed and executed study.

Validity of the findings

This is a qualitative study and so it should be measured against a different set of criteria. The findings and conclusions (once my comments are considered) would be appropriate for the study design.

Additional comments

This is a very important topic. I commend the authors for conducting qualitative research.

Annotated reviews are not available for download in order to protect the identity of reviewers who chose to remain anonymous.

Reviewer 2 ·

Basic reporting

The study provided interesting qualitative materials about HIV disclosure to children among HIV-positive parents and HIV-positive and HIV-negative children in Kenya, which may contribute to studies and interventions about disclosing to children (both HIV-positive or negative) their parents’ HIV status and their own HIV status. This is a well-organized manuscript, but need more work in the part of introduction, for example, the literature review of HIV disclosure could be more sufficient if the authors include several recent literature reviews on parental disclosure and theoretical models of disclosure.
Qiao S, Li X, & Stanton B (2013) Disclosure of parental HIV infection to children: A systematic review of global literature. AIDS Behavior. 17(1): 369-389.
Obermeyer, C.M., Baijal, P., & Pegurri, E. (2011).Facilitating HIV disclosure across diverse settings: A review. American Journal of Public Health, 101(6):1011-1023.
Chaudoir SR and Fisher JD (2010) The disclosure process model: understanding disclosure decision-making and post-disclosure outcomes among people living with a concealable stigmatized identity. Psychol bull. 136(2): 236-256.
Qiao S, Li X, & Stanton B (2013) Theoretical models of parental HIV disclosure: A critical review. AIDS Care. 25(3): 326-336.

Experimental design

Some concerns about the part of methods
1) HIV patients may stand at the vulnerable position in front of their health care providers, are there any alternative procedure rather than standardized one to protect HIV patients during obtaining written consents?
2) Talking about the disclosure experience may result in emotionally uncomfortable (The author mentioned that one kid stopped the interview early and receive psychological counseling), the author may mention what kind of psychological service they could provide in the part of method.
3) Please clarify who conducted the interviews
4) In the last paragraph of the section “data collection and instruments” , the author mentioned “questionnaires” . We usually use “questionnaires” to collect quantitative data, so it may not be appropriate to use this term given the qualitative nature of this study. Either revise it or provide further clarification.
5)How to assess the accuracy of transcripts? Error rates? And when you sent the transcripts to participants, were they randomly selected?

Validity of the findings

1) The sample size is relatively small, which will limit the generalization of the results.
2) The majority of parents are women, gender may play a critical role in their experiences and perceptions, the author may discuss this point.
3) Ensure the quotes are closely related to your statements, for example, one statement “parents whose older children had received full disclosure poorly were hesitant to disclose to younger siblings” is not related to the first quote following.
4) Discussions need to be written in a concise way, without too much repeating results.
5) Although the author explained the participants were not parent-child dyads because of the aims to obtain more diverse data, it may be a limitation for the studies, since we won’t know potential discrepancy between parents and kids in terms of their perceptions and feelings about the same event. The author may discuss it in the part of limitations.
6) There may be too many tables and too many columns for table 1. The basic demographic information (e.g., employment, education, religion, age at diagnosis) may not be necessary unless they are compared with general HIV-positive parents to inform the representation of the sample or are used to identify sub-groups with disclosure patterns (to show the relations between demographic characteristics and disclosure experience).

Additional comments

Language and formats need improved, for example, the font size is inconsistent through the whole manuscript. Some sentences are confusing, e.g. (the last sentence of the limitation part)
“Children and parents are highly sexual-related programs to counteract early sex initiation with the aim of …”

---

## Round 0.2 · Minor Revisions

Your paper has been examined by one of the previous reviewers and we both feel with just a few minor edits, it will be ready for publication. See this reviewers' comments below. Note the suggestion to trim the discussion substantially. I agree with this recommendation as well.

Reviewer 1 ·

Basic reporting

This article meets basic reporting

Experimental design

This is a qualitative study.

Validity of the findings

The study meets the guidelines

Additional comments

Nice job editing and addressing comments. Minor changes to be made:

Methods
Under data collection and instruments delete "subsequently identified as"

Results
2nd paragraph -- that information can be added to Figure 1 and most of that paragraph can be deleted


Discussion
Usually begins with summary of key findings. I would delete first paragraph
Edit the discussion so that it is about 1/3rd shorter.

Delete table 3 (and references to it) or have that as a supplementw

---

## Round 0.3 · accepted · Accept

Thanks for the quick and thorough response to the last round of reviews.